# An Overview of the Origins and Effectiveness of Commercial Fitness Equipment and Sectoral Corporate Settings: A Critical Review of Literature

**Silvio Addolorato** [1,*][ID]**, Jerónimo García-Fernández** [2][ID]**, Leonor Gallardo** [1] **and Jorge García-Unanue** [1][ID]

1   IGOID Research Group, University of Castilla-La Mancha, Avda. Carlos III, s/n, 45071 Toledo, Spain; leonor.gallardo@uclm.es (L.G.); jorge.garciaunanue@uclm.es (J.G.-U.)
2   Physical Education and Sport Department, University of Seville, C/ Pirotecnia, s/n, 41013 Seville, Spain; jeronimo@us.es
*   Correspondence: silvio.addolorato@gmail.com

**Abstract:** Research Question: Fitness equipment is a worldwide ever-growing phenomenon and its usage is nowadays popular both in human routines and academic investigations. Research Methods: This paper is a literature review aiming fitness equipment in relation to all the available findings connected to the complete product life-cycle phases. Results and Findings: Manufacturing industries, which are active realities of the sector, have not been a major concern for sport researchers within the production applicability sub-field. Past root hypotheses, the current state of the art and future guideline applications are addressed. Selected articles were categorised chronologically, by journal, by geographic area and, extensively, by content. Five thematic areas were included: (1) historical background, (2) creation stages, (3) product features, (4) innovation paths and (5) sectoral environments and marketing processes. Implications: By means of the provided findings, there is an opportunity to widen approaches to study fitness equipment that could be extended to the sector's enterprise applications and methods of work.

**Keywords:** fitness equipment; fitness industry; life-cycle approach; literature review; manufacturing industries; product design; product development

## 1. Background

Fitness is generally seen as a late 20th-century phenomenon, but it is important to recognise that some of its roots had already evolved more than a century before [1]. Modern technological fitness equipment has been transformed into a considerable and successful industry, and marketing products and their omnipresence, as human-related machines, play a crucial role [2]. A modern fitness centre is highly efficient in its use of time and space, both in architecture as well as in the simultaneous use of fitness technologies. Users have learned to monitor their own bodies according to the standards of being 'fit'. The globalisation of modern fitness culture can be considered an illuminating example of the more embracing momentum toward a global monoculture, and it is a recognisable and adaptable phenomenon worldwide [3]. The unification and standardisation of fitness (both equipments and practices) have been crucial for the globalisation of the fitness industry [4]. The historical development of modern gym and fitness culture can be described analytically to understand the emergence of this multi-billion-dollar phenomenon [5,6].

The private sector of the global health club industry experiences continuous growth, and sectoral facilities serve 162 million members worldwide [2]. The outlook of the fitness industry is promising

and the sector is expected to thrive in the global marketplace, serving consumers with a variety of health and fitness needs. With access to fitness amenities, instructors, personal trainers and coaches and club operators are well positioned to lead a healthier world.

'Health' and 'fitness' (H&F) are by no means unambiguous terms. The World Earth Organization defined health as a state of complete physical, mental and social well-being, not merely the absence of diseases or infirmities [7]. The term fitness poses similar problems; today it is often equated with muscle size, body contour and the ability to sustain a 30-min exercise bout. It is expected that typical lifestyles of the current and future populations will be opulent and comfortable, as quality of life improves due to the increase in household income and reduction in working hours [8]. In the meantime, as the standard of living becomes increasingly more comfortable, the toll on physical health becomes magnified as a result of bodily weight issues and insufficient exercise caused by super nutrition and change in work conditions (from physical to mental labour).

For this reason, H&F industry is an important contributor to every national preventative health policies against overweight and obesity lifestyles, and directly associated with the related human risk factors [9]. For this reason, every fitness centre plays an important role and, every type of available fitness amenity, is the common mean through which is possible to meet the market demand [10].

This has created a deep awareness of fitness for many people, forcing them to recognise the importance of daily exercise and physical activity. The high annual growth rate in the fitness and athletic equipment market, which is more than 20%, is attributed to this phenomenon [11]. Fitness equipment can be used to improve cardio-pulmonary function, strengthen leg muscles, consume body fat and improve the physical constitution.

It is difficult to make a clear distinction between fitness 'goods' and 'services', since classes or programmes are hardly ever sold without some material goods and vice versa [12]. Fitness products mean equipments, classes, programmes and services (incorporating intangible and new 'concepts') involving all kinds of fitness accessories and amenities, including guide books, DVDs and tutorial videos [12]. Evaluating the fitness world, it has been found as a literal sense of a bodily state, in how various attributes of different extraction come to the intelligible as inhabitable worlds [13].

Fitness equipment is an important part of fitness 'routines'. It has become an unescapable trend due to the rapid development of information technology (IT) and electronic technology, using, for example, photoelectric methods to detect the physiology index of the human body [14]. According to Dibble [15], cardio-vascular equipments are more prone to becoming monotonous, muscle-conditioning devices develop themselves into computerised version of strength training machines. Both categories provide users visual and sonar feedback (based on velocity, measuring distance, range of motion, etc.) bring these 'mature' products to a new innovative era as well.

As globally delivered, H&F services will become more common in the future [4]. The question of standardising bodily movements thus becomes more relevant and could represent new ways in which human immaterial resources are capitalised. Gymnastics, without material equipment, were defined as too routine, and training with new devices generates strong and bold people [16].

Yet, comparatively speaking, researchers in the field of sports product innovation have largely ignored the importance of manufacturing industries as the source of every apparatus available in every sport facility around the world. The fitness 'brand' is independent of the actual product and is unique to the company to which a potential buyer maintains loyalty [17]. For the same reason, during purchasing choices, sport clubs need to take into consideration several aspects and not only focus on supply management of fitness equipment allocation in relation to industry providers [18].

Against this backdrop, the aim of this paper is to provide a traditional literature analysis on all products' life-cycle steps: from the historical hypotheses that lead to the initial sectoral milestones, through the state of the art related to the creation ways of works and the quality variables that compose what already exists in international markets, to the innovation processes that currently define and present this industry's final 'material' results inside contexts and societies.

For this reason, this research is able to highlight this gap in the knowledge about fitness equipment and what has emerged in published studies in relation to sectoral 'companies', simultaneously to all which is defined in literature as product 'featuring', 'innovation' and 'technology' development. The selected objective could also provide and bring practitioners to a newly version of 'win-win solution' between academics and corporate settings, in matching and share useful information. The article presents relevant articles published in the last 40 years, the most important age of the fitness industry. In this context, it has been debated whether sports scientists need to enter this fitness industry sub-field more proactively.

## 2. Methodology

Traditional literature analysis is defined as a method involving a comprehensive search for relevant knowledge and studies on a specific topic, and those identified and selected are then appraised and synthesised [19,20]. The purpose of this type of review is to analyse a large and varied body of literature, in order to understand the current state of the art, in this case regarding -fitness equipment- in relation to other specific independent variables, through an appropriate search in electronic databases.

For this review, the search was not limited to sports-related journals but included articles located in the areas of communication, design, engineering, history, IT, management, marketing, medicine and surgery, psychology and human resources (HR), robotics and aerospace and social sciences. The investigation was limited to research articles in a strict sense, including literature reviews, meta-analyses and original papers. The following typologies were excluded (alphabetical order): bibliographic articles and chapters, books, conference abstract, editorial and news articles, manufacturer or corporative catalogues, sectoral letters and extended product's trademarks (deposited licenses and patents).

A key problem encountered during the analysis is that, while there are numerous corporate case studies and internal enterprise surveys (most unpublished or not made public), little is known about the information available in scientific databases.

The analysis started in November 2015, and the latest search was performed on 31 May 2017. Data collection was performed by one team of investigators experienced in the management and marketing of products in the H&F sector and sports science.

### 2.1. Data Sources and Searches

Electronic databases, including Sport Discus, Academic Search Complete, MEDLINE, ISI - Web of Science, CINAHL, Cochrane Library, PsycINFO, Ergonomic Abstracts, Business Source Premier, Science Direct, Scopus and Google Scholar, were used for the extended literature analysis. This multiple search strategy has been considered as an appropriate strategy due to the eclectic nature and wide range of the targeted literature base (Table 1).

- Searches included the following terms: 'fitness equipment' and (1) 'company', (2) 'feature', (3) 'innovation' and (4) 'technology'. The first key word has been kept fixed varying the remaining four by using the Boolean Logic 'AND'.
- The references of each selected article were evaluated to identify other potentially relevant papers that were not included among the indexed databases utilised by using the known 'snowball' technique [21,22].
- Bibliographies from the retrieved literature were searched, together with the researcher's personal files, to add articles related to the purpose of the research.
- The searches were limited to articles published in English but were not limited by country of origin.
- Relevant industry reports, offered by Europe Active [3] or IHRSA [2,6], were sought by tracking the latest leading trends and geographical/society's product applications.

**Table 1.** Traditional literature review criteria and approaches.

| Criteria | Approaches Adopted |
|---|---|
| *Main topic* | Fitness Equipment |
| *Related topics ('AND')* | (1) Company, (2) Feature, (3) Innovation and (4) Technology |
| *Research method* | Snowball technique |
| *Boundaries defined* | Title, abstract or key words include topics (first phase) |
| | Full-text content articles (second phase, assessed by Jadad scale) |
| | Not by investigation areas, not by period/time frame |
| | Publications in English |

## 2.2. Study Selection and Methodological Quality Assessment

The titles and abstracts obtained were screened to remove irrelevant or duplicated publications. The full-text versions of the remaining papers were then read, analysed and evaluated in detail to identify their eligibility.

The inclusion criteria of the selected articles were as follows:

- Publications related to the content criteria for 'fitness equipment' that were published through academic journals and directly intersected with the topics of 'company', 'feature', 'innovation' and 'technology' that should appear in the title, abstract, or keywords.
- Where aims were not included in those topics, it has been analysed if has been extensively faced or treated between the offered contents of the full-text version.
- Articles published between 1978 (the lower limit was not previously defined at the beginning of the review) and 2017 at the date of the last search performed.
- Papers written in English.

Once the relevant publications were finalised, the Jadad scale [22,23], was used to systematically determine the quality of the articles for approval and acceptance. Having been done entirely by a single reviser, a high standard of the results was sought for analysis selection. The standard set for selection was all the articles that earned a punctuation of three or more points, which indicates good quality. The chosen scale has the capacity to provide a complete overview of the external and internal validities of papers included in this revision to determine the current state of the art.

Being a traditional literature review (and not a systematic one), the process through which these twelve scientific selected databases have been analysed corresponds to the initial sample of material retrieved. The final search, defined by the selection and quality assessments above listed, revealed 73 articles, of which 50 were found applying the previously explained content-related criteria.

## 2.3. Data Analysis

The rationale for this approach was that, targeted thematic analysis has been considered a useful and flexible tool, which can potentially provide a rich and detailed understanding of eclectic data [19]. Additionally, this method enables researchers to identify, analyse and report patterns of meaning (themes) across different epistemological and ontological positions.

When the selection was ready, findings were organised into five key paragraphs following the traditional product's steps development [24]: (a) historical background, (b) creation stages, (c) product features, (d) innovation paths and (e) sectoral environments and marketing processes.

This overall categorization has been defined according to the logical life-cycle processes associated with the fitness equipment (especially from the point 'b' to 'e'), with a starting addition (point 'a') focused on the origins and roots (formally named as 'milestones') of the commercial fitness equipment provided by the selected paper during the analysis.

## 3. Results and Discussion

When considering the years of publication, an evident increase in papers on the focused topics of interest was observed. Until the beginning of the 21st century, there is no continuity on production. From 1978 (lower limit defined during the analysis) to 2006, only 14 articles were published (28%). In the last 10 years (from 2007 to 2016), a mean of more than three publications per year (M = 3.1) was evidenced, with two peaks during 2011 and 2014 with a mean of 5.5 selected articles (Figure 1).

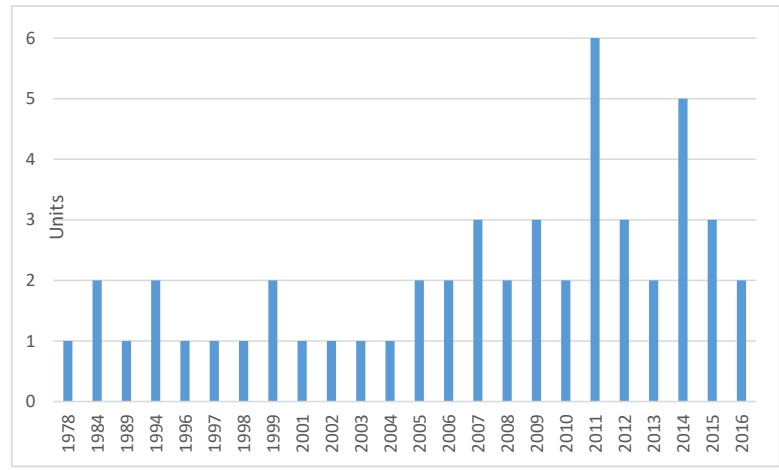

**Figure 1.** Number of publications by year.

Regarding the origin of the published papers, the results evidenced a wide diversity of interest areas, as previously listed. Up to 22 journals have been published on the topics investigated. Journals with a greater number of publications were as follows: International Journal of the History of Sport with three articles and Applied Ergonomics, Journal of Sport Management and Managing Leisure (now Managing Sport and Leisure) with two papers each. These nine publications represent more than a third (40.90%) of the journals on the subject studied (Table 2).

**Table 2.** Journal titles and number of related publications.

| Journal Title | Number of Publications per Journal |
| --- | :---: |
| The International Journal of the History of Sport | 3 |
| Applied Ergonomics<br>Journal of Sport Management<br>Managing Leisure (now: Managing Sport and Leisure) | 2 |
| ACSM's Health & Fitness Journal<br>Acta Astronautica<br>Current Sports Medicine Reports<br>European Business Review<br>European Journal of Cultural Studies<br>European Journal of Marketing<br>Expert Systems with Applications<br>Gender, Place and Culture: A Journal of Feminist Geography<br>Human Resource Management Journal<br>International Journal of Precision Engineering and Manufacturing<br>International Journal of Sports Marketing and Sponsorship<br>Journal of Beijing Sport University<br>Journal of Sport History<br>Perceptual and Motor Skills<br>Presence: Teleoperators and Virtual Environments<br>Scientia Iranica<br>Social Marketing Quarterly<br>Sport Science Review | 1 |

A geographical analysis may also be as enlightening following some sample studies encountered in literature, mostly focused on European [25] and North American [26] environments and their revenues, cash flows and profitability.

Regarding the geographical distribution from which the selected studies have been performed (both first author and co-author), the most active academics have been in Europe with 41.57% (22 articles) of the total sample assessed, followed by the Asia-Pacific area, with 33.96% (n = 18) and North America, with 20.75% (n = 11). The lowest representation was in the Middle East, with 3.77% (n = 2). No publication matched the established criteria for Africa (including an association for the Middle East and North Africa (MENA) area, as unified by the descriptors for the northern part of the continent) or Latin America (Table 3).

**Table 3.** Numbers and percentages of publications' geographic areas.

| Geographic Areas | Countries | Selected Articles | |
|---|---|---|---|
| | | *N* | % |
| Asia—Pacific | Australia | 1 | 33.96 |
| | China | 9 | |
| | Japan | 1 | |
| | South Korea | 3 | |
| | Taiwan | 4 | |
| Europe | Denmark | 1 | 41.57 |
| | Finland | 3 | |
| | France | 2 | |
| | Germany | 2 | |
| | Ireland | 1 | |
| | Norway | 2 | |
| | Sweden | 2 | |
| | The Netherlands | 1 | |
| | UK | 8 | |
| Middle East | Jordan | 1 | 3.77 |
| | UAE | 1 | |
| North America | US | 11 | 20.75 |

The European robust performance, but above all evidently spread in all the continent and confirmed by sectoral investigation, led in first position by the UK with annual revenue of 6.1 billion dollars/year [2,6]. Even in the Asia-Pacific area there is a correlation with the investigation's findings; these countries are home to maturing industries and current growth, and main examples are found in cities such as Beijing, Shanghai, Kuala Lumpur and Jakarta. For North America, this analysis was represented exclusively by the US; the latest annual revenue obtained was 27.6 billion dollars/year, accounting for more than 35,000 facilities among all states [2]. No criteria matched studies from Canada. Despite conflicts in several MENA countries, there is a demand for fitness, as consumers seek to exercise and reap the benefits of an active lifestyle. Recently, successful international fitness operators have expanded their operations into this area.

### 3.1. Historical Background

Some investigations encountered through this state of the art analysis, revealed the historical root phases on what defined the fitness equipment categories arriving until how are nowadays internationally labelled. These works came mainly from other literature reviews and are here redacted with a chronology logic for two reasons: (a) to introduce the following paragraphs (product's life-cycle processes) and (b) to provide the current and updated vision of the fitness equipment sector since its inception (Appendix C).

Van Hilvoorde [1], using a continental European approach, explains that a French visionary named Hyppolite Triat, in the late-19th century, was one of the first to develop fitness 'ideals' by opening commercially-exploited gymnasiums in Liège, Brussels and Paris. Gymnastics were transformed into theatrical spectacles, planned with the circus groundwork (trapezes, beams, rope ladder and masts), and a complete collection of weights (dumbbells, barbells, Indian clubs, etc.). At the same time, in the Dutch context, Dr. Geraldus Arnoldus Nicolaas Allebe has been cited for his greater relevance. In Germany, evidence has been found of ideas and machines focused on isolated muscular groups but not developed within educational contexts. In sum, in this stage, fitness equipment was 'freed' from ideological associations and was neutral with regard to variables such as gender, age and social class (public or private spaces). During the same historical moment, when the first 'artist-entrepreneurs' such as MacFadden and Sandow finally arrive to the final audience, the evolution of fitness devices in journals, for sports and physical education have definitively started. In this period, the main communication tools were newspaper advertising, periodicals, books, photography and films.

In the late-19th century and the first three decades of the 20th century in Germany, England and North America, distinctions between weight-lifting, bodybuilding, wrestling and circus-strongman performances were not always clearly drawn [27]. These various disciplines were neither considered nor practiced separately, and they were sometimes even understood as an organic whole. The final goal in training with weights was to increase strength and develop an attractive body. These numerous ideas and models were brought together around the turn of the century under the general term of 'Life Reform Movement'. In these environments, the most utilised means of advertisement were records, lectures, specialist magazines and exercise handbooks.

In the past century, between the sixties and seventies, a number of events affected studies about the history of medicine, the body, health and fitness [7]. The 'motor' metaphor was a useful device for studying the body's physiological and neurological processes. Some of the most useful works concerning H&F were focused on exercise, physical training, sports and athletics. During the same period, a Chinese perspective was provided by Zhang [28] that divided the national sports industry into three stages: exploratory (1978–1992), formative (1993–1996) and developmental (1997-present).

The extrapolated studies also focus on 'Nordic' milestones of the fitness industry, from a wider range of view, from the beginning [16] to the latter half of the 20th century [29]. The first study addresses the introduction of gymnastics as a subject in the Norwegian school system. The rooms allocation and the choice of equipments were used as instruments to form pupils' bodies and minds. The equipments already included were: climbing ribs (less than 100-cm-wide sections), ropes, rope ladders, ladders, net ladders, beams, vaulting horses, box horses, bucks, springboards, cushions, balls, jump ropes and mats. In contrast to Central Europe provided version [1], these equipments must never transform gymnastics into acrobatics and 'circum-like' conditions. The second publication [29] considerable attention to the development and organisation of fitness exercise inside the Scandinavian Peninsula (Denmark, Norway and Sweden). This contribution brings to the actualised vision of the activity forms included in for-profit fitness in three main types: (1) individual training studios, with a variety of physical-amenities, aimed at strengthening specific body segments; (2) group-based trainings (aerobic cardio-works, etc.) focused on cardiorespiratory endurance; and (3) relaxation exercises, not always human-product related, that include a variety of wellness concepts and approaches. These historical roots and milestones bring fitness equipment to the 'modern era' of the fitness industry.

### 3.2. Creation Stages

The first step, in the modern age of the fitness industry is, undoubtedly, given by the phases needed to generate, from a cero or basic level, a new piece of training system, or almost define the creation stages in what already exists in the current society (Appendices A and B).

There is evidence that multi-sectorial teamwork is at the foundation of daily life manufacturing industries [30]. Biomedical mechanics, manufacturing and design engineers, IT specialists and human body practitioners act in unison. Countries around the world typically divide special

workout equipments into four creation categories: (1) hydraulic, (2) electric/electromagnetic damping, (3) pneumatic and (4) gravity-mechanical. Always during the concept creation, in relation to the procedures through which is normally to operate inside an enterprise, another classification of products is provided by Fu [14] that divide equipment between oxygen and aerobic. Currently, it is hard to meet the fitness needs of modern people with 'traditional' apparatus so both domestic and international entities have begun to explore more 'intelligent' fitness equipment. These categories of machines can collect physiological parameters and providing a wide variety of related functions (virtual channels, explanatory videos, etc.), measure the size of ejection forces, and provide quantified training useful for both trainers and trainees [30].

Products typically follow a cycle of introduction, growth, maturity, decline and termination [31]. The H&F industry is in the early maturity phase of the sector life-cycle. In this regard, Williams et al. [32] underline that it is not always easy, depending on the magnitude of the reality in question, to acquire 'Top of Mind Awareness' (TOMA) in potential future customers. There are three primary reasons why brand awareness is a component of brand 'equity': to increase customer consideration, create effective decision-making and influence the brand association's (product and/or service) formation.

As affirmed by two studies [33,34], manufacturing industries also need to take into account variables related to the final operators, users and facilities in which they, regularly, are purchased. Between these, there are safety practices connected to the hardware's layouts, with operating procedures and conditions required to correctly apply products in health centres. The four overall factors included in the analysis are the enterprise's guidelines for safe operations: (1) the importance of spaces for hypothetical hazards, (2) possible barriers for people with physical disabilities, (3) obstacles or stairs to 'get inside' the machine and (4) the challenge of creating specific new fitness equipment (supposedly the hardest topic) that does not contradict the previous points.

Current recommendations affirm that fitness gyms and private health clubs are an outstanding global business, but new 'trends' such as franchising chains, and specific-environment workout spaces are the new frontiers of product creation aimed at final purchase and consolidation [5].

### 3.3. Product Features

The literature suggests a direct connection among all variables that define, catalogue and divide fitness equipments in the electronic databases in the past decades. Since the beginning of the 21st century, it has become evident that there are few differences among the main bodies of the products (indexed to each single sub-category), brands and marketing promotions [35]. The barriers of entering and withdrawing in the trades become bigger, and the point of view of the industrial organisation (from theory to the practice) plays a key role. The size and mass of the devices are important parameters [36], as well as other parameters analysed in the literature (Appendices A and B).

Numerous studies focus on the indispensable physiological, anthropometric and biomechanical paradigms [37]. H&F devices have wide application: laboratories, training centres, competitive contexts, etc. These are the reasons why these body-related factors are defined as basic knowledge in the field. The versatility, for a product designed with ergonomic criteria, indicates hypothetically high levels of usability. As seen in the visionary approach of Reilly and Thomas [38], the man-machine interface needs anthropometric data to give correct responses to the users who interface with it. Physiological methods and time-motion analysis have also been evaluated as relevant variables. Starting from ergonomic reasons, and still related to multifunctional applicability, the degrees of freedom of the amenity is defined as important including the rehabilitation active phases [39].

Similarly, for biomechanics, flexibility concerns studies do not directly relate to the material sciences, but rather to the adaptable meaning of the word. Scientific evidence underlines the importance of its effects and the ability to be used in various contexts [1,40]. Still, muscle strengthening, actions (concentric, eccentric, etc.), reactions, the variety of speeds of contractions and muscle recruitments and all the relations with bone concepts have been assessed, through selected devices, by various authors reviewed [36–38,40,41]. The initial comparison is often related to exercises carried out with free

weights and, as a more generic definition provided, the workout against supra-normal loads. Findings reveal that non-electrically powered, yet 'gravity-independent' equipment, has similar features to 'traditional' equipment known to be effective in producing the desired physiological responses [41].

In relation to the previous feature listed, the quality of usability is mentioned in some studies [29, 33,38,42,43]. The main topics highlight the importance of the final user guidelines and response types, depending on the type of customer (lately including senior categories) and the operative capacities when a monitor or display is included in the physical amenity. Other quality variables treated include the educational aspects for H&F professionals, addressing, for example, the application variables needed to make the most of the selected products from each manufacturer in their own context (overall population, population per km$^2$, area in km$^2$, penetration rates, etc.).

Five additional tangible quality parameters have been mentioned in most evaluated publications in relation to existing products: (1) attractiveness [4,38], (2) efficiency [17,43], (3) high quality and experienced design control [1,4,42], (4) functional and practical linearity [1,38] and (5) specialty and cost [30]. Additionally, three intangible aspects emerged: (1) beneficial [17], (2) engagement and (3) 'magic moments' during every use of the fitness apparatus [4].

The analysis continues with the environmental parameters in which the products need to be placed and begins with a source evaluation [36]. The surrounding structure or the external environment is not affected through unnecessary noise or induced vibrations, especially when the evaluated equipment does not have a mechanical system included in the hardware parts. These aspects could indicate the insecurity of the product and cause subordinated issues related to the increase of temperature, humidity and lighting vibrations [42].

Aside from being one of the most discussed topics, sectoral providers already know the importance of the overall safety and hazard prevention related to the physicality of the equipments which is not fundamental only for the health care and rehabilitation applicabilities, but also for the entire population who daily uses the products [44]. Many studies address this topic to the machines final features [1,17,30,33,38,39,43]. Evaluations began from preventive acts to avoid hurts and joint injuries during utilisation, until touch the comfort in creating a safe fitness 'experience'.

### 3.4. Innovation Paths

It is widely accepted that creating quality products and services require a well-trained and skilled workforce, business strategies and polyfunctional abilities that give momentum to the fitness industry innovation [45]. The factor for successful innovation is not exactly the same for every company [46]. Sorting and analysing the individual cases can serve as a basis of reference and act as a guidepost for other companies in similar industries during their business development. Subsequently, it is possible to strengthen the enterprise's capacity in order to accelerate process innovation, reduce dependence on other economies and develop marketing brand strategies to promote and generate more added values [28].

To create an attractive fitness brand, an enterprise does not necessarily need innovation products or services, but something, or someone, with personality and the right preparation to give vitality to the brand [12]. Since quality of life is rising constantly, the fitness industry has developed quickly due to the high demand worldwide [47]. A lack of innovation, in addition to other related factors, such as product research and technical development, could restrict the number of members which participate to the delivered course of service practices [48]. The innovation activity of a sectoral trader (procedure inputs/processes outputs) is separated into three phases: motivation, process and performance [46]. Through these stages, the enterprise's competitive advantages and efficiencies can be clearly seen.

Thirty years ago, Dibble [15] made an observation that represents the current reality and could be written today with an eye to the future (p. 74):

> *"The fitness industry is far from exhausted [ . . . ] manufacturers seem taking a rational and scientific approach in designing the next generation of fitness equipment [ . . . ] not satisfied with simple muscular or cardiovascular improvements they are taking steps toward an intelligent workout that*

*optimises user's time [ . . . ] another design target is increased interaction between user and the equipment [ . . . ] if you can 'see' progresses and intensities you will be more inclined to provide an extra-effort required to correctly complete an exercise".*

Studies have explored seven quality innovation areas desired for a product's overall growth, which should be kept in mind: (1) bodily and physiological, (2) tangible corporeality, (3) intangible corporeality, (4) intangible variables, (5) trade actions, (6) field personnel and (7) final consumer (Appendices A and B).

These seven key points, divided along the continuation sub-paragraphs, correspond to the main fitness equipment facets obtained from the traditional literature analysis performed (product life-cycle phases and manufacturing key players).

1    Bodily and Physiological

The first quality dimension of H&F equipments is directly related to the human body and its physiological parameters. These are useful to entirely embrace all that is required to carry out physical activities through physical amenities.

As noted by Reilly and Lees [37], training apparatus has a similar emphasis on exercise specificity, but not all of its related functionalities are triggered by the required physiological movements. Another analysed theme is the standardisation of body gestures and all the named kinaesthetic skills [4]. Additionally, the structure needed to meet the ergonomic requirements is planned in order to render the patterns more comfortable for different groups of people who will use the equipment [47]. Between the safety qualities, connected to biomechanics, there are the related and required input-admission measurements [11], and body dimensions data useful to categorise the user type [42]. The physiological changes induced by physical activity, such as motion sickness, spatial disorientation, orthostatic hypotension, muscle atrophy and bone demineralisation, are evaluated by Davis and Davis [36].

2    Tangible corporeality

Continuing to analyse the tangible qualities, required to create new products, is normal to list all the attributes that appear to the insiders manufacturers view and, as well, to the end-users perception.

Everything that is made under the innovative constructs begins from physical security and comfort directives [37]. Material science has contributed, and will continue to do so, on a large scale in hitting implements. In addition, strict safety factors are highly demanded at all buyer levels [42]. Safety guarantees in the design should decrease accident probabilities. Additionally, the morphologic semantics shape has been studied. The overall size and structure of the equipments have also been analysed to create a wider applicability [47].

3    Intangible corporeality

Another important innovation aspect related to corporeality is everything immaterial that can be directly handled by manufacturers and play a vital role for industry operators and, especially, end-users. What works best for the success of a product is the innovative design perspective of hardware and software amenities [17,37]. For example, an experienced design of the fitness services brings to the key factor of the 'lived' body [4]. The logo design of the company has also been included among the evaluated 'attributes' and obtained a rating of 5.83 ± 1.02 on a 7-point Likert scale for purchase decisions [49].

Among the four main factors of industry success there is the 'mastering technologies' facet [46]. In this regard, companies gain a great chance to extend their business by paying more attention to technological innovations [47]. Combining concepts of modern technology (personal computers, tablets, etc.) a device is normally perceived closer to the user's ways of work [11].

Only one study considered the order of methods to conduct energy storage [11], and another study aimed on how IT visions are transmitted vertically and horizontally to the global market [4].

Effects of virtual reality and interactive interfaces have been studied in the user view for adherence, and to obtain real-time exercise information to check the accuracy [47]. Involvement and adherence have been assessed by two research groups [50,51], where a virtual reality enhancement exercise obtained the highest levels of attendance and effects of sensory input, such as music and video feedback, have been studied by means of computerised attentional focuses (gaze strategy orientations).

## 4　Intangible variables

All those 'sensations' produced by the equipment that are reflected on fitness professionals and, especially, to the final consumers are an important sphere of innovation which is also covered by all those non-material-themed aspects (Appendix B).

During their leisure time (when most use fitness equipment), users want to avoid boredom, time complaints, and more in general need something enjoyment to not impend in reasons of declining interest [15,51]. This is often correlated to studies of the areas of emotional and sensational knowledges and body-related skills [4].

Various psychological variables have been evaluated in relation to fitness devices [49,50,52]. Among these are positive engagement, revitalisation, tranquillity, physical exhaustion, dissociation, regulated competition, enjoyable experience and all the 'benefits' (identification, acceptance, nostalgia, place pride, etc.). The only article addressing the psychological suggestions related to the product's colours is offered by Wang and Gao [42].

## 5　Trade actions

The final applicability and adaptability of fitness equipment in the environments and international communities represent the triumph or not of the same. As affirmed by Guest and Taylor [53] regarding product orientation, success comes to those organisations that offer goods and services considered 'good' by the final audience. Efficiency of corporate costs, efficient production (phase cycles) and optimal distribution systems and channels are at the foundation of sectoral affirmation.

In competitive sport activity contexts, it is already known that bilateral correlation between specific disciplines and related equipment is not always clear [37]. Thus, product's marketing actions must be analysed and defined in terms of similarity (or differences) from other objects that might occupy the same environment in relation to the sector competitors [12,17]. Additionally, still connected to the machinery's trading, have been assessed corporate revenues and profitability for each created and moved equipment inside targeted environments [54].

Among the additional guidelines emerged in literature, three are cited: (1) create a clear brand position [46] including themed area contents in relation to health and rehab applications in order to obtain more hypothetical active consumers [11]; (2) differentiate products and be a smart marketer [44]; and (3) study and focus on 'basic' application variables such as geographies, cultures, genders, races and classes [55]. The costs of new and innovative product or service development in the global market have risen, prohibiting the emergence of new brands [12].

## 6　Field personnel

Behind every device there is a defined multi-area skill force of practitioners, most of them starting from the employment of diversified talents [44,46].

According to Guest and Taylor [53], excluding the assembly, IT, design and engineering departments, a total of 56% of corporate respondents said that their principal professional background was in sports and leisure management, followed by administrative or general management at 16%. For the fitness industry, there is no clear link between competitive strategies, and established skills [45]. Among the attributes of the sectoral brand equity there is research evaluating the role of 'star players' within companies [49].

Two enterprise case studies provide new approaches ways to develop innovative ideas. Kennedy-Armbruster et al. [17] explain that there is a need for collaborative education models (translational education) between institutions of higher education and for-profit corporations to create

real personal relationships. Another contribution, facing the company's personal accountability and ownership, established a direct connection between employees and sectoral trader's financial performances [54]. The OZ Principle® (Getting Results Through Individual and Organisational Accountability) encompasses four steps: see, own, solve and do it.

7   Final consumer

After the conception, creation, features applicability and, where possible, innovation, every product needs an audience by which public demand is generated. Indeed, the ultimate judge of the adequacy of public sector leisure provision is the customer [53].

Field consumer-based brand equity is defined through 16 reviewed dimensions divided by attributes, benefits and acceptance [49]. Yet, it is equally true that in the way to know one's potential consumer, a clear winning choice should be that the companies could study the final consumer before [44]. Contemporary societies do not matter the criteria and paths in which human-machinic fitness products are nowadays assembled [55].

Enterprises are looking for the real consumer's feedback from all points of view [17]. Generate maximal attendance and create or increase levels of interaction are the main objectives at this level of the negotiations [15,50,51].

*3.5. Sectoral Environments and Marketing Processes*

An important step is to define what has emerged, from the systematic searches in relation to the final environments in which the featured products normally arrive, and which marketing strategies have already been studied and performed. These steps logically occur after the creation, featuring and innovation stages.

The first topic includes the relation between H&F experts and employees, and final consumers [56]. The study affirms that social marketing approaches in this direction have grown rapidly within the past 20 years. Aside from the normal skills required by the sectoral staff, nowadays interest in self-development is needed to provide better and competent services in relation to what is physically owned. Healey and Marches [57] observe that several practitioners are also being trained in disease prevention, motivational techniques and health care marketing skills to motivate the same employees, users and their health-related concerns. The fulcrum of marketing models involves the consumer orientation in product/service development, which is why a wellness program is usually critical in its first step regardless of the environment in which it is implemented. According to Grönroos [58], the quality of the services, physical or otherwise, is divided into 'expected' (ad's, field selling, public relations, pricing, traditions, word-of-mouth, etc.) and 'perceived' (in which is included the cover image and connect technical/functional qualities).

The final usage has also been assessed, taking into consideration the financial and access performances and their relationships [59]. A relevant consumer point of view is provided by McKechnie and colleagues [60], who analysed the female market segment (aimed mainly to home fitness devices), as a viable target for ads claims through in-depth interviews and a specific questionnaire. The information sought included the type and brand of the equipment in possession, which family figure bought the device and the reasons that bring to this decision, and, finally, what opined on the future development of this targeted kind of product. Another investigation interviewed sales managers of a studied area and acknowledged that customers rarely complain about the overall qualities of the purchased devices. The types of products included in the analysis were treadmills, cycling machines and workout (muscle strength) apparatuses, including small abdominal-focus amenities. Almost half of the final consumer cohort (44%) affirm that believe in ads claims, and, of those, 34% confirmed that their belief in ads messages depends on the origin of the ad. Always facing the sale moments of the product processes, You et al. [18] identified two main phases: the equipment allocation decisions and the algorithm that determines the sales limits (expected sales numbers for consumers and management in terms of member recruit limits).

The relationship between a brand's association and loyalty was studied by Gladden and Funk [52] in relation to professional sports consumers. Results showed 'positive' relations with fan identification, escape, nostalgia, product delivery and loyalty; 'negative' with tradition, star player(s) and peer group acceptance. The commercial sponsorship approach has been analysed in terms of its impact on purchasing products in relation to a brand's image and attachment [61]. Based on the nature of the entities involved, the model demonstrates that multiple sponsorships activate three 'brand behavioural dimensions': cognitive, affective and conative. The changes of post-industrial environments, with a focus on marketing brands, embrace various other influences [62]: local traditional value views, production and distribution globalisation aspects, hierarchical system transition (e.g., when one brand become sub-brand), the transformation development of family-run management and practices of an enterprise's liabilities.

Finally, facing the strategic part of the H&F marketing actions has evaluated the demand compliance and the supply-led and interactive convergences investigating the interconnection between the usage of fitness equipment and the related manufacturing industries [63]. Findings show that production establishments were the Granger cause of training on fitness products and, where not, the opposite relation was confirmed only for supply-led convergence. Nowadays, in this stage of the industry, fitness providers predict future development tendencies through the main competitors' product analysis, the innovation provided and price variations [64]. Marketing includes: culture, service, brand allocations and all these variables mixed together. Economic indications are dictated mainly by social development states of international negotiations within fitness industry.

## 4. Conclusions

This paper has synthesised the ways in which fitness equipment has been studied in relation to the life-cycle phases of the product-amenities introduction. Since the beginning of the historical and visionary milestone ideas, passing through the concrete creation of companies focused on production, until the ultimate trends that, this modern industry, is nowadays still able to offer. Additional findings demonstrate how the sectoral enterprise's path is far from finalised. The goal is still to continuously introduce innovative routes to train and approach people with new human-machine physical activities.

This traditional review of the available research indicates that there is a literature gap in the way of 'white label' investigations that not merely need to mention, in a specific manner, companies or their related products. These actions are more common for the parallel physiological research areas, applied to physical exercise sciences and everything that is instrumental or diagnostic, which are conducted on fitness, or rehab tools, mandatorily named along the scientific manuscripts. Similar speech could be done for the same corporate realities that are barely interested in bringing their internal researches to academic scopes, and limit their own marketing analysis to private procedures, aimed only at 'defending' themselves from external/peer competition.

The five key points identified do not stand alone because an inter-relationship is evident that combines each of these steps. For the historical roots is undoubted that, the encountered nods, could represent only partially the complete atmosphere that stood from the late-19th century and the 20th-21st centuries. The opinions included have been the only that match the criteria initially established for the literature analysis. Going forward, the review illustrates that, regardless of the production areas analysed, several quality product features correspond with each other. For this reason, it is possible to confirm that the guidelines provided by the literature (and for this, is supposed, that have been already followed by the named manufacturing realities) could be considered valid and, moreover, viable in the coming decades given the increase in demand/purchase actions of fitness equipments at international levels.

Firstly, this review indicates that local support, which is already in the possession of the final investors and users, is fundamental. This needs to be 'ready to use' and applicable for every consumer group, or professional, who needs to interface with the amenity and deserves to enclose the right balance between corporeality and the accompanying software services (where included). Secondly,

understanding the evident lack of specific-themed equipments applicable to each sport discipline outside the fitness area, the current study argues that there is a necessity to develop products that could be more useful (and permit these customers to feel more the inclusiveness) for this population of already focalised athletic individuals (specific movements, fewer hardware barriers, wearable connectors, etc.).

The geographic areas that did not emerge in the review, or at least were less cited by the selected sources, could correspond to where less products are entirely or partially developed. No academic publications are available in supporting this quote, but within the sector it is well known that the fitness industry leaders are located mainly in North America and Europe [6]. At the same time, this may inform understanding that, with the advent of the new century, companies also locate their headquarters in countries with potential internal development (e.g., Far East, Middle East, Latin America), and not only for classic reasons of cushioning costs in producing, rather than in places labelled as more saturated. Academic contributions are, for this reason, waited in the future also by these 'fresh' fitness environments. This could mean taking into account a mix of cultural, economic, political and social aspects that shape these focused processes.

It is argued that pure management and marketing approaches are still weak for those study areas not included in home fitness equipment or hospitalities (hotels, etc.). These 'extra slices' are still at the foundation of labour-routine of sectoral sales force (B2B level), but are less assessed in academic literature. Interesting variables could be the range of products available, geographical distribution and its related information, and details intersected to the disposable society to which actions are directed. This could provide information on additional variants that could be investigated to why a specific amenity is considered suitable for a determined market segment or not.

*Limitations and Recommendations*

Authors acknowledge both strengths and limitations to the approach adopted for reviewing and analysing the existing evidence.

The main strength from this traditional literature analysis is the level of flexibility offered by our approach. This enabled us to include a wide variety of information without being impeded by strict or inappropriate inclusion/exclusion criteria. Moreover, the iterative approach allowed us to refine the analysis based on newly discovered material.

Among the limitations of this review, the initial problem was the large number of published licenses and patents related to the products registered worldwide that also appear in academic databases and were discarded because of the content's irrelevance. In addition, secondary searches were carried out, according to the same evaluation methods with the common synonyms used in the publications for the key words selected (provided in alphabetical order):

*Equipment*: accessory, device, good, instrument, item, hardware/service amenity, machinery, machine, object, quality physical material, product, system, tool, etc.

*Company*: brand, concern, corporation, enterprise, entity, industry, provider, sectoral trader, etc.

The subjectivity of the authors was inherent within the process. This approach means that the review's quality depends upon the review team's skills, like in other methods of investigation. The main ability has been weave together the relevant material in a systematic and logical way. Nevertheless, we believe that the adopted approach is the most suitable way to fulfil the purpose of this research given the current state of the sub-field analysed, as well as for the volume of material included in the investigation. A final limit is the number of databases used and the keywords selected, which could have been expanded to include more scientific supports.

Further investigation approaches could focus their analyses on a specific product-phases or, at the same time, on how various steps, highlighted in this contribution, are correlated to those which are definable as more proximal (e.g., 'featuring' with 'creation' -pre- and 'innovation' -post-), with all the includable scientific variables assessable. Or, maybe, include only a specific type of the most commercial fitness equipment models (treadmills, strength-isotonic amenities, etc.) available in a

specific market segment, or simply dictated by the trend of the moment in which the analyses are carried out.

'Theoretical to practical' implications, which could be directly transposed to the current fitness industry and its players at all levels (manufacturers, B2B, B2C, etc.), correspond to the outcomes retrieved in each part of the human-machine product life-cycle assessment obtained: from the design, stepping through the overall featuring development, until the effective innovation provided in all material and non-material results which reaches the correspondent sectoral population of 'devotees'. Especially for this last process of field renovation, the polyhedral approach in terms of themes sorted demonstrates that this industry need a wide variety of practitioners which could work at unison in the same direction.

Acknowledging all the inter-relationships between the stages needed to produce fitness equipment and all the applications carried out by the related manufacturing companies will be an important first step for those who want to understand the complexity of this interesting fitness industry sub-sector for further academic investigations.

**Author Contributions:** Conceptualization, S.A.; Formal analysis, S.A.; Investigation, S.A.; Methodology, S.A.; Project administration, S.A.; Supervision, L.G.; Validation, J.G.-F. and J.G.-U.; Visualization, J.G.-F. and J.G.-U.; Writing—original draft, S.A.; Writing—review & editing, S.A. All authors have read and agreed to the published version of the manuscript.

**Funding:** This research received no external funding.

**Conflicts of Interest:** The authors declare no conflict of interest.

## Appendix A

### *Resume of Selected Articles on Specific Product Features and Qualities*

| Product Features and Qualities | Author/s (Year) |
|---|---|
| *Action flexibility* | Berg and Tesch (1998); Davis and Davis (2012); Kang et al. (2013), Reilly and Lees (1984), Reilly and Thomas (1978), van Hilvoorde (2008) |
| *Anthropometric, biomechanical and physiological parameters* | Davis and Davis (2012), Kreighbaum and Smith (1996), Lee (2008), Long et al. (2009), Parviainen (2011a), Reilly and Lees (1978), Reilly and Thomas (1984), Teng et al. (2014), Wang and Su (2012), Wang and Gao (2013) |
| *Applicability and usability* | Burton and Huffman (2007), Kreighbaum and Smith (1996), McComack (1999), Reilly and Thomas (1984), Steen-Johnsen and Kirkegaard (2010), Wang and Gao (2013), Wang and Su (2012), Yoon and Ahn (2015) |
| *Brand (attachment, equity, position and vitality)* | Gladden and Funk (2001, 2002), Chavanat et al. (2009), Parviainen (2011b), Ren (2007), Williams et al. (2014), Yuan et al. (2009) |
| *Color* | Wang and Gao (2013) |
| *Corporate educational models* | Kennedy-Armbruster et al. (2011), Takaki (2005) |
| *Corporate revenue and profitability* | Takaki (2005) |
| *Design control* | Gladden and Funk (2002), Kennedy-Armbruster et al. (2011), Kreighbaum and Smith (1996), McKechnie et al. (2007), Parviainen (2011a), Reilly and Lees (1984), Wang and Gao (2013) |
| *Economical specialty* | Teng et al. (2014) |
| *Emotional and sensational knowledge* | Parviainen (2011a) |

| IT and virtual reality | Yuan et al. (2009), Long et al. (2009), Wang and Su (2012), Parviainen (2011a) |
|---|---|
| Linearity | Reilly and Thomas (1978), van Hilvoorde (2008) |
| Safety and comfort practices | Burton and Huffman (2007), Kennedy-Armbruster et al. (2011), Lee (2008), Parrot (1996), Reilly and Lees (1984), Reilly and Thomas (1978), Sekendiz et al. (2016), Teng et al. (2014), van Hilvoorde (2008), Wang and Gao (2013), Yoon and Ahn (2015) |
| Size and mass (structure) | Davis and Davis (2012), Long et al. (2009) |
| Source type and energy storage | Davis and Davis (2012), Wang and Su (2012) |
| Temperature, humidity and light | Wang and Gao (2013) |

## Appendix B

*Resume of Selected Articles on Induced and Transmitted User's Effects*

| User Effects (Induced or Transmitted) | Author/s (Year) |
|---|---|
| Attractivity | Parviainen (2011a), Reilly and Thomas (1978) |
| Beneficial | Kennedy-Armbruster et al. (2011) |
| Efficiency | Kennedy-Armbruster et al. (2011), Yoon and Ahn (2015) |
| Engagement | Annesi and Mazas (1997), Parviainen (2011a) |
| Enjoyment | Dibble (1989), Mestre et al. (2011) |
| Interaction (on attendance) | Annesi and Mazas (1997), Dibble (1989), Mestre et al. (2011) |
| Involvement (on adherence) | Annesi and Mazas (1997), Mestre et al. (2011) |
| Physical exhaustion, regulated competition and revitalization | Annesi and Mazas (1997) |

## Appendix C

*Resume of Selected Articles on Named Companies (or Services) on Geographic Basis*

| Authors (Year) | Country | Named Company/Ies or Workout Concepts |
|---|---|---|
| *Annesi and Mazas (1997)* | US | DiamondBack Fitness, Tectrix |
| *Berg and Tesch (1998)* | Sweden | Flywheel |
| *Burton and Huffman (2007)* | US | Concept2, Cybex International Inc., FitLinxx, Life Fitness (Brunswick Corp.) |
| *Dai (2014)* | Japan | Asics, Mizuno |
| *Dibble (1989)* | US | Belton, Cybex International Inc., Hydra-Fitness (Hydra-Gym Athletics), Life Fitness (Brunswick Corp.), Nautilus Inc., Precor Inc., Reebok, StairMaster (Randal Sport/Medical), Universal |
| *Kang et al. (2013)* | South Korea | Biodex Inc. |
| *Kennedy-Armbruster et al. (2011)* | US | Johnson Health Tech. Co., Ltd. (Matrix branch) |
| *McCormack (1999)* | UK | NordicTrack |
| *Parviainen (2011a)* | Finland | Les Mills Int. |
| *Parviainen (2011b)* | Finland | Putkisto's Method (stretching, pilates and face school) |
| *Reilly and Lees (1984)* | UK | Cybex International Inc. |
| *Sekendiz et al. (2016)* | Australia | Cybex International Inc., Johnson Health Tech. Co., Ltd. (Matrix branch), Life Fitness (Brunswick Corp.), Nautilus Inc. |
| *Takaki (2005)* | US | Precor Inc. |
| *Yuan et al. (2009)* | Taiwan | Johnson Health Tech. Co., Ltd. (general) |

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
