# Peer review of "An Overview of the Origins and Effectiveness of Commercial Fitness Equipment and Sectoral Corporate Settings: A Critical Review of Literature"

_applsci, doi:10.3390/app10041534_

Round 1

Reviewer 1 Report

This Article is overall well researched and must be published because completely matches the aims of this special issue on Design, Manufacture and Analysis of Sports Equipment. I find it a very interesting Paper in which the work objectives entirely correspond to the Results (with Discussion) and Conclusions. This Work aims to investigate all the life-cycle product approach landscapes in Fitness Equipment research which has been underrepresented so far.
It is important to emphasize that these kind of deep and systematic literature reviews are a very important tool to base the work in certain areas of knowledge, and that they are sometimes little used, especially in this investigation field. I think that with this very appropriate Methodology, more work that follows this line should be carried out.

- Keyword: is better do define “literature review”, instead of “analysis”. Include the keyword "fitness industry".

- Background, this paragraph is very well-written contextualizing the present Article’s objectives within preceding research landscapes in the target domain

- Methodology. In the second paragraph “HR” appears. It does not describe what it means.

There is an error in the word "sectoral." Do the authors refer to “sectorial”? It is written in different paragraphs of the article.

There is an error in some cites and references. Please include citations and references where "Author A" appears.

Please state the following statement: Relevant industry reports, offered by Europe Active (2018) or IHRSA (2017; 2018), were sought by tracking the latest leading trends and geographical / society’s product applications.

Has this methodology been used in other studies? Please cite and reference this methodology (see Table 1).

Selected papers appear in results. These data should appear in the methodology, in particular in subsection 2.2.

The practical implications of the study are not clear.

Reviewer 2 Report

It's quite an extensive manuscript on an issue that is not often the subject of scientific considerations. Therefore, it is interesting for a rather narrow group of researchers dealing with issues of well being, public health, fitness, etc. How can this thesis be confirmed by the fact that during the examined period of 1978-2017 the authors gained access to and analysed only 73 articles. At the same time, from the scientific point of view, the article presents high quality. The way of describing the methodology indicates a high level of research workshop. The results were combined with a discussion, which brings the text to life. The way of interpreting the results does not raise any objections, the authors have grouped the analyzed issues, which increases the usefulness and gives the article more practical value. The chapter "limitations and recommendations" can be considered as a strong point of the article.
